

# Transient LES of an offshore wind turbine

Lukas Vollmer, Gerald Steinfeld, and Martin Kühn

ForWind, Carl von Ossietzky Universität Oldenburg, Institute of Physics, Küpkersweg 70, 26129 Oldenburg, Germany.

*Correspondence to:* Lukas Vollmer (lukas.vollmer@uni-oldenburg.de)

**Abstract.** The estimation of the cost of energy of offshore wind farms has a high uncertainty, which is partly due to the lacking accuracy of information on wind conditions and wake losses inside of the farm. Wake models that aim on reducing the uncertainty by modeling the wake interaction of turbines for various wind conditions need to be validated with measurement data before they can be considered as a reliable estimator. A methodology is shown to create realistic transient wind conditions in a Large-Eddy-Simulation of a marine boundary layer interacting with an offshore wind turbine for a direct comparison of modeled with measured flow data. A mesoscale simulation is used for determining the boundary conditions of the model. The simulations of the ambient wind conditions and the wake simulation generally show a good agreement with measurements from a met mast and lidar measurements, respectively. Advanced metrics to describe the wake shape and development are derived from simulations and measurements but a quantitative comparison is difficult due to the scarcity and the low sampling rate of the available measurement data. The methodology presents a possibility to compare flow measurements with simulations. Due to the implementation of changing wind conditions in the LES it could be also beneficial for case studies of wind turbine and wind farm control.

## 1 Introduction

Offshore wind energy still remains an expensive alternative of electric power generation compared to onshore wind energy which has established as one of the cheapest options to generate electricity. One of the reasons for the comparatively high costs of offshore wind energy is the scarcity of atmospheric measurements at existing or planned wind farms. The resource assessment at these locations is difficult and prone to large errors (Walker et al., 2016). In addition, missing measurements during operation prohibit the thorough analysis of turbine malfunctions and unexpected underperformance.

Only few offshore wind farms deploy a permanent met mast that allows to study the influence of atmospheric conditions on wind farm performance. Due to the lower level of turbulent kinetic energy offshore, compared to onshore, the wakes of the wind turbines are more persistent, which leads to higher wake losses at downwind turbines even over larger distances. An even lower turbulence level caused by stable atmospheric stratification leads to a further increase of wake losses (Barthelmie and Jensen, 2010; Hansen et al., 2012; Dörenkämper et al., 2014).

Several numerical models have been developed with the purpose to calculate the optimal layout of offshore wind farms under consideration of the wake losses. Engineering models allow a fast calculation of multiple wind scenarios and an optimization





of wind farm layouts (Sanderse et al., 2011). These steady state models however have a low representation of the flow physics and rely largely on the parametrization of turbulence and on a simplified interaction of turbine and wind.

A high fidelity solution for wind farm simulations are Large-Eddy-Simulations (LES). Coupled with wind turbine models, LES provide a detailed solution of the flow inside of a wind farm with a high representation of the relevant physics. Due
to the high computational costs, LES of offshore wind farms have yet been restricted to exemplary simulations of idealized atmospheric conditions or to case studies of specific situations, e.g. Churchfield et al. (2012); Dörenkämper et al. (2015); Lu and Porté-Agel (2011).

An issue of all wind farm models is the permanent need for validation with measured data to evaluate the capability of the model of reproducing the actual wind conditions and performance of the wind farm under these conditions (Moriarty et al.,
2014). Besides performance measurements from the data acquisition system of wind turbines, that are often confidential, flow measurements using the light detection and ranging methodology (lidar) have become a popular tool for scientific research. To optimize this technology for model validation, the lidar measurement campaigns have to be designed and postprocessed to allow for a fair comparison with simulations. One aspect of the measurement design is the measurement of free flow conditions which can be used as meteorological boundary conditions for the simulations.

Especially offshore the measurement or derivation of boundary conditions to set up the simulations is challenging. For example, onshore LES are often run with boundary conditions derived from near-surface measurements (e.g. heat flux measurements) and are compared to wind profiles derived from lidar devices (Mirocha et al., 2015; Machefaux et al., 2015) or met masts. This procedure is rarely possible at offshore sites because usually near-surface measurements and wind speed profiles are not available. Furthermore, for a lot of models additional input is required, e.g. the height of the atmospheric boundary
layer or a large scale pressure gradient to drive the flow. These properties are rarely measured and have to be estimated or set to default values.

In this paper we investigate a methodology to use profiles and boundary conditions derived from a mesoscale simulation for a continuous LES of an offshore wind turbine wake over multiple hours. The purpose is a direct validation of the flow distortion by the wind turbine model with the wind field extracted from lidar measurements (van Dooren et al., 2016) during
the simulated timeframe. The use of the mesoscale profiles allows us to include synoptic-scale meteorological conditions. Recently, long term LES of multiple days up to one year have been run with this approach to study the changes of meteorological conditions at a measurement site (Neggers et al., 2012; Schalkwijk et al., 2015; Heinze et al., 2016). In the context of wind energy the approach was yet only used in Vollmer et al. (2015), where the measurement and simulation setup of this paper was briefly introduced. Here we extend the study in Vollmer et al. (2015) by analyzing a much larger time interval of measurements
and simulations, by a sensitivity study of the method and by a quantitative comparison of wake characteristics.

The content of the paper is structured as follows: After introducing the measurements and the model equations, we evaluate how different setups of the LES model and different input from the mesoscale model contribute to the LES capability to simulate the measured state of the atmosphere. For the most suitable setup, a simulation was run with high resolution and a wind turbine model of the turbine, for which lidar measurements of the wake flow were available. Different measures are compared




to evaluate the similarity of simulated and measured wake. We conclude how the combination of forcing with a mesoscale simulation and the use of a wind turbine model is able to simulate the measured wind and wake conditions.

## 2   Data and methodology

### 2.1   Measurement data

The case study that is analyzed in this paper is based on measurements on 20 February, 2014 at the German offshore wind farm *alpha ventus*. Two independent data sets are used for comparison with the model results. The simulated ambient wind conditions without turbine are compared to measurements from the met mast FINO1 located at N $54° 01'$, E $6° 35'$. Time series from cup anemometers, wind vanes and temperature probes at different heights as well as surface temperature from a buoy are provided by the Bundesamt für Seeschifffahrt und Hydrographie (BSH). These time series provide mean values obtained from

averaging over ten minutes. The wind directions of the wind vanes at all heights were corrected using a direction-dependent bias (DEWI, personal communication). During the analyzed timeframe the flow measurement devices do neither operate in the mast shadow nor in the wake of a wind turbine, thus should provide high accuracy information of the marine atmospheric boundary layer.

The lidar measurements, that are used for comparison with the simulated wakes, are part of a measurement campaign that

took place from August 2013 until March 2014. During the analyzed day two long range lidar devices (Windcube WLS200S) executed single elevation PPI scans in the wake of turbine AV10 with one lidar positioned on FINO1 and the other one on the converter station of the wind farm (Fig. 1).

The line of sight velocities of the lidars were combined and averaged to get a ten-minute mean horizontal vector wind field at hub height (van Dooren et al., 2016). Measurements were filtered at both ends of the range of the Carrier-to-Noise level to

remove low backscatter data as well as reflections from objects. Wind fields that do not contain enough measurements within a reasonable altitude range centered around hub height and timeframes in which a lot of yaw activity of the turbine was observed were removed from the database. Averaging was done on volumes with a diameter of $20\,\mathrm{m}$ centered at hub height. Because both lidars scan over a relatively small range of azimuth angles (Fig. 1), seven (WLS2) and five (WLS3) sweeps over the scan area contribute to the calculation of the mean velocities. The view of the lidar devices to certain areas of the scan is blocked

by other wind turbines, thus a varying total of 100 - 350 individual line-of-sight wind speed values contribute to the average at each grid point of the final wind field. The coordinate system, in which the flow field is presented, is oriented north by scanning the distance to the turbines of the wind farm with known geographical coordinate positions. More information on the calculation of the vector field from the line of sight velocities can be found in van Dooren et al. (2016).

### 2.2   Model equations

Revision 1928 of the LES model PALM (Maronga et al., 2015) is used for this study with the same numerical schemes as in Vollmer et al. (2016). The extension of the model equations to include time-dependent forcing is based on Heinze et al.





(2016) with an extension to include large scale advection of momentum. The modified equation of motion before applying any approximations is:

$$
\begin{aligned}
\frac{\partial u_i}{\partial t} = &\underbrace{-u_j \frac{\partial u_i}{\partial x_j}}_{1} \underbrace{-\epsilon_{ijk} f_j u_k}_{2} \underbrace{-\frac{1}{\rho} \frac{\partial p}{\partial x_i}}_{3} \\
&+ \underbrace{\nu_m \left( \frac{\partial^2 u_i}{\partial x_j^2} + \frac{1}{3} \frac{\partial}{\partial x_i} \frac{\partial u_j}{\partial x_j} \right)}_{4} \underbrace{- \epsilon_{i3j} f_3 u_{g_j}|_{LS}}_{5} \\
&+ \underbrace{\frac{\partial u_i}{\partial t}\Big|_{LS}}_{6} \underbrace{- \frac{<u_i> - u_{i_{LS}}}{\tau}}_{7}
\end{aligned}
\tag{1}
$$

with term 1 representing the momentum advection, term 2 the Coriolis force with the Coriolis parameter $f_j$, term 3 the pressure gradient and term 4 the friction terms with the kinematic viscosity of momentum $\nu_m$. Terms 5-7 are the external forcing terms. For the external forcing a large scale velocity denoted by $|_{LS}$ is defined. Term 5 defines a large scale pressure gradient by prescribing a horizontal geostrophic wind speed $u_g$. Term 6 prescribes a large scale sink or source of momentum and term 7 is a time relaxation of the momentum towards a large scale state (Neggers et al., 2012; Heinze et al., 2016).

The momentum relaxation has no physical justification but is used to prevent a drift of the model from the large scale state. The term depends on the difference between the horizontal average $<u_i>$ of each velocity component and the large scale velocity component $u_{i_{LS}}$, scaled by a relaxation time constant of $\tau$. All large scale properties only vary on the vertical axis $x_3$.

The equation for scalars $s \in (\Theta, q)$ is:

$$
\frac{\partial s}{\partial t} = \underbrace{-u_j \frac{\partial s}{\partial x_j}}_{1} + \underbrace{\nu_s \frac{\partial^2 s}{\partial x_j^2}}_{4} + \underbrace{\frac{\partial s}{\partial t}\Big|_{LS}}_{6} \underbrace{- \frac{<s> - s_{LS}}{\tau}}_{7} + \underbrace{S_s}_{8}
\tag{2}
$$

with terms 1, 4, 6 and 7 equivalent to the corresponding terms in eq. 1, with $\nu_s$ being the molecular diffusivity of the scalar. Term 8 is the surface flux of either the specific humidity q or the potential temperature $\Theta$.

Time-dependency of the external forcing is achieved by prescribing profiles of the time-variant geostrophic wind, source terms of horizontal momentum and of scalar properties, as well as of the large scale state of the relaxation term. The surface fluxes are calculated by making use of the Monin-Obhukov similarity theory, with the values of the surface pressure, temperature and humidity also prescribed by the time-dependent large scale state.

## 3   Simulation of free stream flow

In this section we analyze the simulation of the ambient conditions with the large scale forcing derived from the output of a mesoscale simulation. Different parameters are modified to analyze their influence on the results. In the first part we look at the meteorological conditions that were present at the day of the measurements. In the second part we compare profiles from the



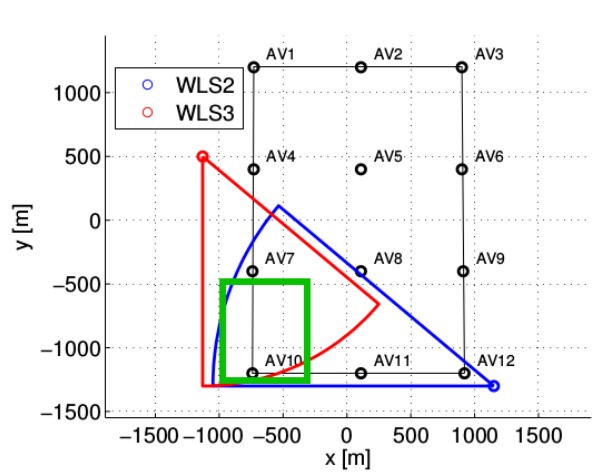

**Figure 1.** Layout of *alpha ventus* and position of the two lidars that were used for the construction of the wind field. Circular segments denote the scan areas of the lidars. The green box denotes the region of the vector wind field reconstruction.

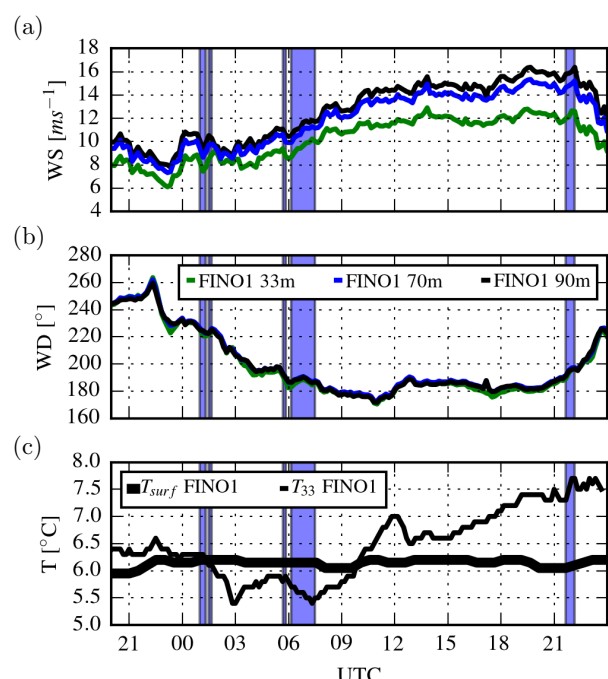

**Figure 2.** Meteorological conditions on 20 February, 2014, as measured at FINO1. (a,b) Wind speed and wind direction at different heights. (c) Temperature as measured at height and temperature of the sea surface. The periods of the selected lidar measurements are shaded in blue.

mesoscale model with the FINO1 measurements. In the third part we compare the LES model output of different setups with the mesoscale model and the FINO1 measurements.

## 3.1 Meteorological conditions

The lidar measurements were conducted on 20 February, 2014. After filtering according to the criteria mentioned in Sec.

5  2.1, 15 ten-minute timeframes remained for further analysis. The 15 timeframes can be sorted into three periods, with three measurements starting at 01:00 UTC (night period), nine around 06:00 UTC (morning period) and another three starting at 21:40 UTC (evening period) (Fig. 2).

The wind direction at FINO1 is south-west during the night and south during the rest of the day with an increase of the wind speed at hub height of the *alpha ventus* wind turbines (90 m) from about 8 ms$^{-1}$ to about 16 ms$^{-1}$. The day is a rather warm

10  winter day, with the measured temperature at 35 m ranging from 5.5 to 8 °C. Compared to onshore the diurnal cycle of surface temperature is very small at offshore locations because of the large heat capacity of the ocean surface. The observed drop of

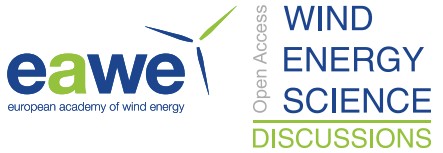

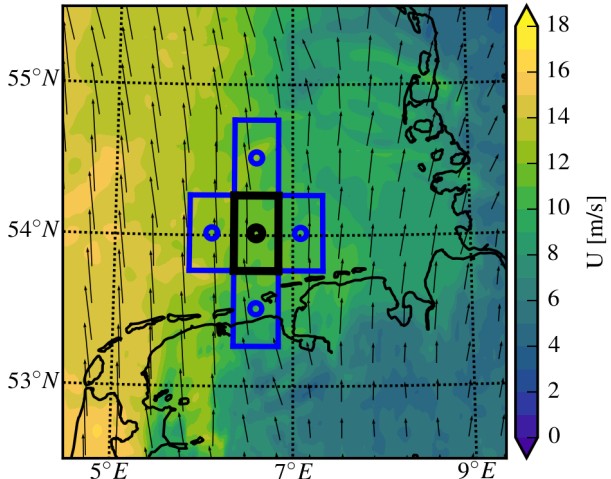

**Figure 3.** COSMO-DE wind speed and direction at 20 February, 2014, 07:00 UTC on the model level of 73.5 m. The black square marks the averaging domain surrounding FINO1 and the blue squares the neighbouring domains that are used for the calculation of the gradients.

air temperature during the morning hours is thus most likely related to the advection of colder air from the land. The German coast is approximately 45 km to the south of *alpha ventus*, thus for a wind speed of 8 ms$^{-1}$ the advection of onshore air masses takes about 1.5 h. The advected cold air leads to thermally slightly unstable conditions between about 2 am and 9 am. During the rest of the day the stratification is slightly stable.

## 3.2   Input data from COSMO-DE

The profiles for the large scale tendencies are calculated from the operational analysis of the COSMO-DE model (Baldauf et al., 2009) of the German Weather Service (DWD). The COSMO-DE model has a horizontal resolution of 2.8 x 2.8 km and 50 vertical levels in total with 20 vertical levels in the lower 3000 m of the atmospheric boundary layer. The DWD delivers
10    hourly model data.

Following Heinze et al. (2016) three-dimensional and surface data is averaged over a horizontal averaging domain of multiple grid points. The nearest grid cell to the FINO1 coordinates is chosen as the center of the averaging domain. Because of the necessary spin up time of the LES for the development of turbulence, 24 hours of spin-up time (February 19, whole day) were added. The profile of the geostrophic wind is calculated using the pressure gradient between neighboring averaging domains
15    (Fig. 3). The component of the geostrophic wind along the west-east axis is defined by:

$$u_{g_1}^{I,J} = -\frac{g f_3}{\rho^{I,J}} \frac{P^{I,J+1} - P^{I,J-1}}{2 dX_2} \tag{3}$$





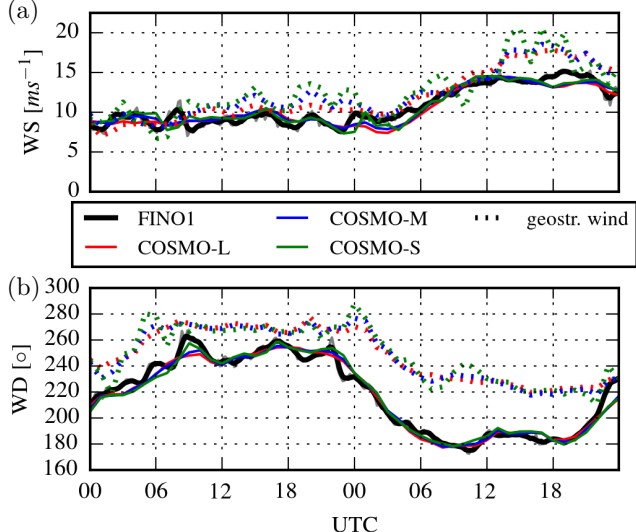

**Figure 4.** Time series for Feb. 19 and 20, 2014 of (a) Wind speed at 70 m. (b) Wind direction at 70 m. FINO1 1-hour running average (black). COSMO-DE averaging domain sizes of $(1/8)^2$ degrees (COSMO-S), $(1/2)^2$ degrees (COSMO-M), $2^2$ degrees (COSMO-L). Dotted lines represent the calculated geostrophic wind speed and direction at the same height.

with $P^{I,J}$ and $\rho^{I,J}$ the domain-averaged quantities of density and pressure in domain (I,J) and $dX_i$ the grid length of the averaging domain. The north-south wind component $u_{g_2}^{I,J}$ is defined accordingly. The source terms $\frac{\partial u_i}{\partial t}|_{LS}$, $\frac{\partial \Theta}{\partial t}|_{LS}$ and $\frac{\partial q}{\partial t}|_{LS}$ result from the advection into the averaging domain, the source term of the potential temperature $\Theta$ for example is defined by:

$$\frac{\partial \Theta}{\partial t}|_{LS} = U_1^{I,J} \left( \Theta^{I+1,J} - \Theta^{I-1,J} \right)$$
$$+ U_2^{I,J} \left( \Theta^{I,J+1} - \Theta^{I,J-1} \right) \tag{4}$$

We analyzed the influence of the size of the averaging domain on the profiles required by the LES model by comparing three different quadratic domain sizes with grid lengths of $1/8$ degree, $1/2$ degree and $2$ degree. Fig. 4 compares the measured 70 m wind speed and direction from FINO1 with the horizontal and the geostrophic wind speed and direction from the different averaging domains.

Fig. 4 shows that the wind speed and wind direction are close to the measurements on both days. As expected, less fluctuation on about an hourly timescale occurs in the mesoscale profiles compared to the measured ones. This can be either related to mesoscale fluctuations that are not reproducible with the model or to the horizontal inhomogeneity of the flow that is measured at FINO1.

The comparison of the different averaging domains shows that the smaller domains contain more fluctuation, but not neces-




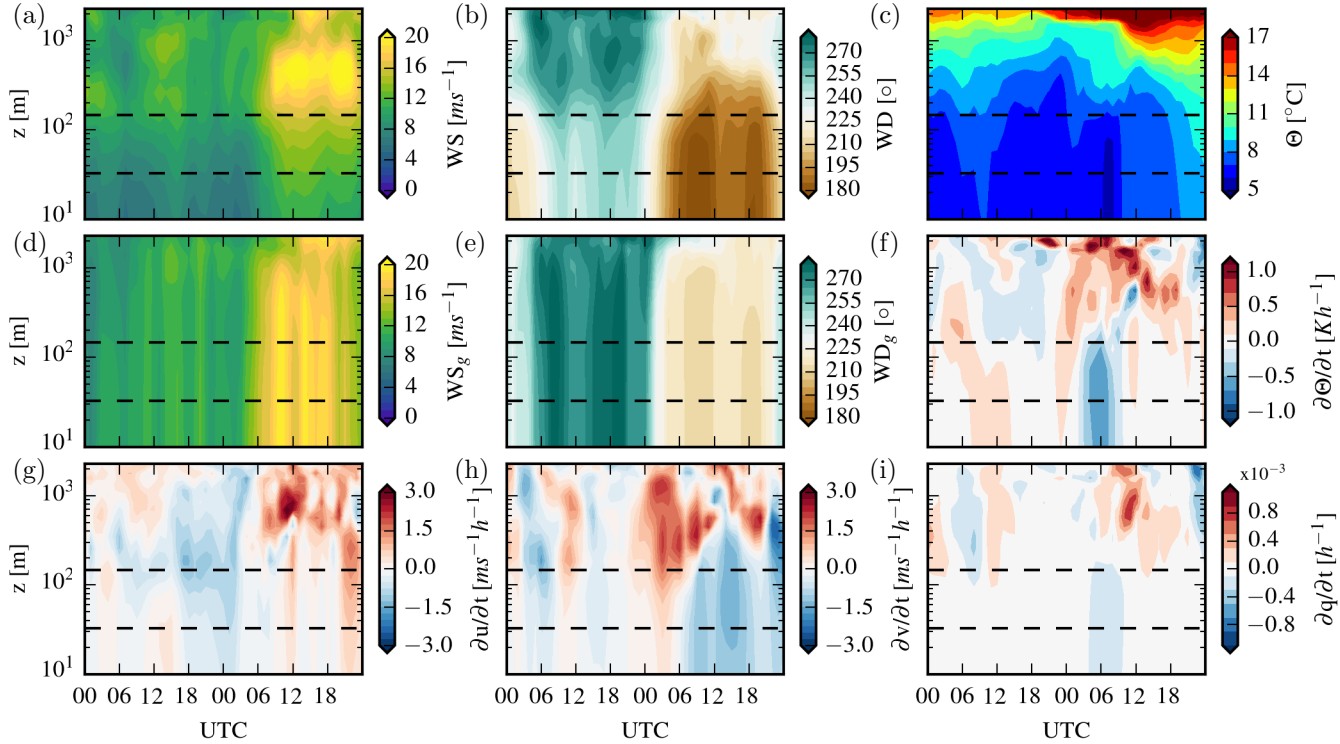

**Figure 5.** Time development of the vertical input profiles for the LES run. (a) Wind speed, (b) wind direction, (c) potential temperature, (d) geostrophic wind speed, (e) geostrophic wind direction, (f) advection of potential temperature, (g) advection of zonal wind speed, (h) advection of meridional wind speed, and (i) advection of humidity. Dashed horizontal lines represent the lower and upper rotor tip height.

sarily at the same time as the measurements. In addition the geostrophic wind that is calculated from the pressure gradients in the model becomes noisier with decreasing averaging domain size (Fig. 4). Because the geostrophic wind is directly used in the equations of motion we chose to use the middle sized domain. It generally contains less noise than the small domain and in contrast to the large domain, it contains just grid points over the sea.

5      Figure 5 shows Hovmöller diagrams of most of the large-scale forcing data we used for the LES model. As assumed earlier, an advection of colder temperature during the morning hours of the second day is visible in the mesoscale simulation (Fig. 5(f)). The change of wind direction with height is mostly related to the Ekman turning, which can be seen in the differences between the geostrophic and the effective wind direction (Fig. 5 (b) and (e),respectively).

### 3.3   Comparison with met mast data

10   To transfer the input profiles from the COSMO-DE time steps and height levels to the LES model, they were linearly interpolated on the vertical LES grid and on the time steps of the simulation. The LES were initialized with the set of large-scale profiles at 19 February, 00:00 UTC and nudging was applied only after six hours to enable a free development of turbulence in



**Table 1.** Comparison of the different simulation setups and the RMSE ($\sigma$) of wind direction and speed to the COSMO-DE input (C), the
FINO1 measurements (F1) and the reference simulation ($P_{ref}$)

| Sim | $\tau$ | $\Delta x, z$ | $\sigma WD_C$ | $\sigma WD_{F1}$ | $\sigma WD_{P_{ref}}$ | $\sigma WS_C$ | $\sigma WS_{FINO1}$ | $\sigma WS_{P_{ref}}$ |
|---|---|---|---|---|---|---|---|---|
| | [h] | [m] | [°] | [°] | [°] | [ms$^{-1}$] | [ms$^{-1}$] | [ms$^{-1}$] |
| $P_{ref}$ | 4 | 20/10 | 5.8 | 6.6 | - | 0.79 | 1.08 | - |
| $P_{\tau_1}$ | 1.5 | 20/10 | 4.0 | 5.1 | 2.5 | 0.73 | 0.99 | 0.16 |
| $P_{\tau_2}$ | 48 | 20/10 | 7.5 | 8.3 | 2.8 | 0.70 | 1.10 | 0.36 |
| $P_{\partial_t u=0}$ | 4 | 20/10 | 8.5 | 9.4 | 6.2 | 1.53 | 1.76 | 1.02 |
| $P_{\partial_t \Theta=0}$ | 4 | 20/10 | 6.9 | 7.6 | 1.8 | 0.77 | 1.08 | 0.14 |
| $P_{hi}$ | 4 | 5/5 | 5.7 | 6.5 | 1.4 | 0.46 | 0.91 | 0.42 |

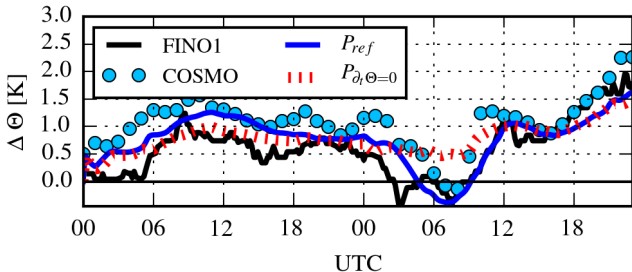

**Figure 6.** Potential temperature difference between 35 m and the surface.

the first hours.

All simulations had a domain size of 3200 m x 3200 m x 1700 m and were run with cyclic boundary conditions. The roughness length of momentum was taken from the COSMO-DE model ($z_0 = 1.23 \cdot 10^{-4}$m), the roughness lengths of temperature and humidity were $z_0^{\Theta,q} = 0.1 z_0$.

Five different simulations with a rather coarse grid were run with different configurations (Table 1). One of the setups was then run with a finer resolution for the turbine simulations. The chosen setup is regarded as the reference simulation $P_{ref}$ and the simulation with higher spatial resolution is called $P_{hi}$. Two alternative relaxation time constants were set in $P_{\tau_1}$ and $P_{\tau_2}$ and advection of either momentum or potential temperature was switched off in $P_{\partial_t u=0}$ and $P_{\partial_t \Theta=0}$.

10 For evaluation we selected the ten-minute mean wind speed at 70 m, as it is close to the hub height wind speed and is also available from the COSMO-DE model. For better comparison the raw ten-minute wind speed of the anemometers was smoothed by means of a one-hour running mean. Table 1 shows the setup of the different simulations and the Root Mean Square Error (RMSE) of the time series of wind speed and direction from the mesoscale model, the measurements, and from





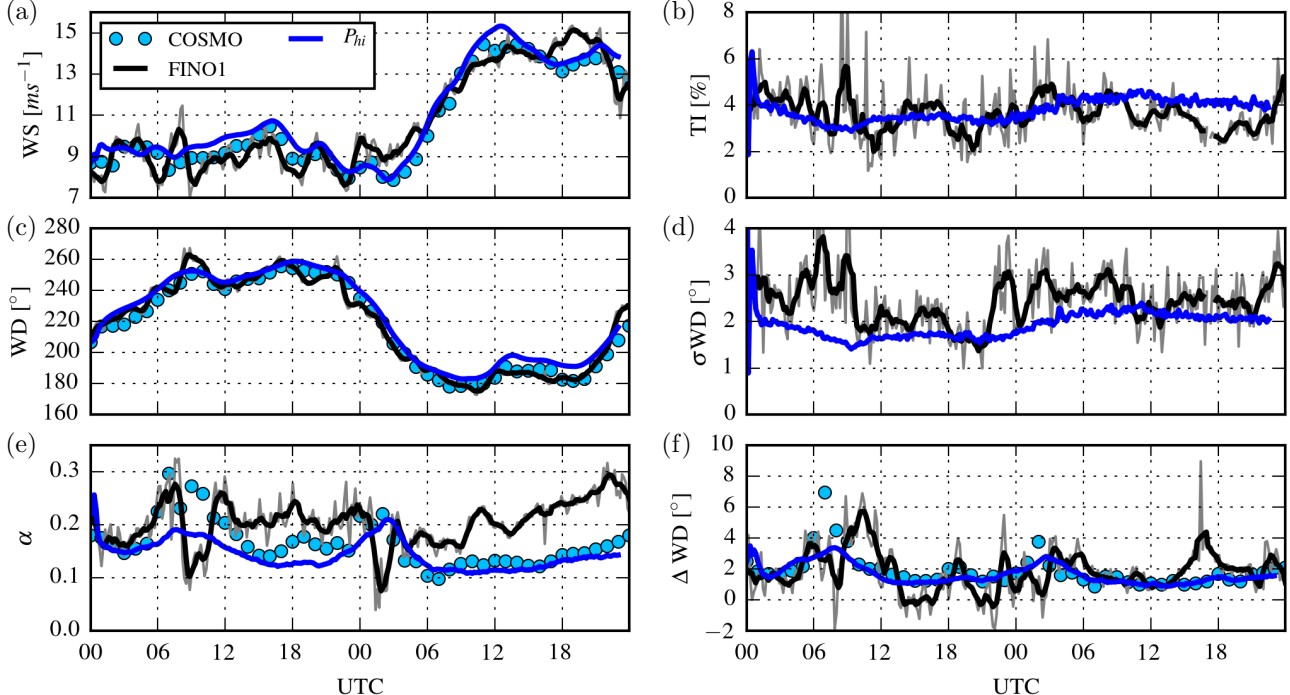

**Figure 7.** Comparison between one hour running means and ten-minute averages (grey) of FINO1 measurements, COSMO-DE and $P_{hi}$. All timeseries at 70 m, if not specified otherwise. (a) wind speed, (b) turbulence intensity, (c) wind direction, (d) ten-minute standard deviation of the wind direction, (e) vertical power law coefficient as defined in the text, and (f) change of the wind direction between 33 m and 70 m.

the domain average of the reference simulation.

As Tab. 1 shows, switching off momentum advection appears to have the largest influence on the wind speed and wind direction deviation from the input data. The advection of potential temperature does not have a large effect on the time series of the wind speed at 70 m. But Fig. 6 shows the importance of the temperature advection for the thermal stability as the reversal of

5 the temperature gradient can indeed be closely related to advection. As the measurements from FINO1 are not directly used for the model setup, it is difficult to really quantify the quality of the different simulations as a smaller RMSE does not necessarily mean that the relevant physical processes are better resolved. For example, the bias of the wind direction which is already provided by the forcing data is not expected to vanish because of the use of LES. Thus, the decision for a relaxation time of 4 h for the turbine simulation was mostly a qualitative decision because a slower relaxation should allow the LES boundary layer

10 to develop more independently.

The timeseries of the domain averaged results of $P_{hi}$ are compared to the measurements and the large-scale forcing data in Fig. 7. The ten-minute turbulence intensity TI and the standard deviation of the wind direction are calculated at virtual met masts inside the model domain and averaged over the values at different positions. The power law coefficient is calculated for the FINO1 measurements and the LES from a fit to the data between 33 and 90 m and for COSMO-DE by using the model





levels at 35 and 73 m.

Wind speed and wind direction of the simulation follow the trend of the input and measurement data. The magnitude of the turbulent fluctuations on the ten-minute scale is well reproduced. Turbulent fluctuation on the scale of about 1-4 h on the other hand are not reproduced with the model, as they are also not part of the mesoscale model that delivers the forcing data.

The largest discrepancy between simulation and measurements exists in the shear of the vertical wind profile which is almost constantly lower in the LES.

The destabilization of the boundary layer is clearly visible in the vertical shear of the LES and the measurements (Fig. 7 (e)) on the second day between 00:00 UTC and 06:00 UTC. The event appears to occur earlier in reality than in the simulations, which is likely related to the earlier change of sign of the temperature gradient (Fig. 6). The restratification also starts later in

the LES and the vertical shear remains constantly lower during the rest of the day.

## 4 Wind turbine wake simulations

### 4.1 Model setup

The wind turbine wake simulations are run with the same domain and setup as the high resolution simulation $P_{hi}$. The wind turbine is placed in the middle of the domain. Due to the cyclic horizontal boundary conditions, the wind turbine wake reenters

the domain through the southern after having left it through the northern boundary of the domain. However, as the wind direction in the simulations is never directly from south, the turbine is not directly in its own wake. The turbulence of the wake still modifies the state of the atmospheric boundary layer in the cyclic domain after some time. Thus, we made simulations without the turbine in parallel and simulated only intervals of 30 min maximum with wind turbine with a 3 min precursor phase for the development of the wake.

An enhanced actuator disc model with rotation (ADM-R) is used to calculate the forces of the wind turbine on the flow (Witha et al., 2014). The model divides the rotor surface into annulus segments, and the local velocities at the segments and tabulated lift and drag coefficients are used to calculate lift and drag forces. Tower and nacelle of the turbine are parameterized by constant drag coefficients. The parameterized wind turbine AV10 is an Adwen AD 5-116 with a rotor diameter of 116 m and a rated power of 5 MW. The hub height of the turbine is at 90 m. Adaptation to the current wind conditions is ensured

by a baseline generator torque and pitch controller as described in Jonkman et al. (2009) and a simple yaw controller. The yaw controller is implemented as described in Storey et al. (2013). with a relatively short time averaging window of the wind direction of 30 s and a tolerated maximum misalignment of 5 degrees.

### 4.2 Comparison with lidar measurements

Figure 8 compares parameters that indicate the state of the atmospheric boundary layer measured at FINO1 with the simulated

state during the selected 15 ten-minute timeframes of the lidar measurements at *alpha ventus*. As already discussed earlier, the biggest disagreement is found in the vertical shear which is constantly lower in the simulations. The TI is slightly higher in



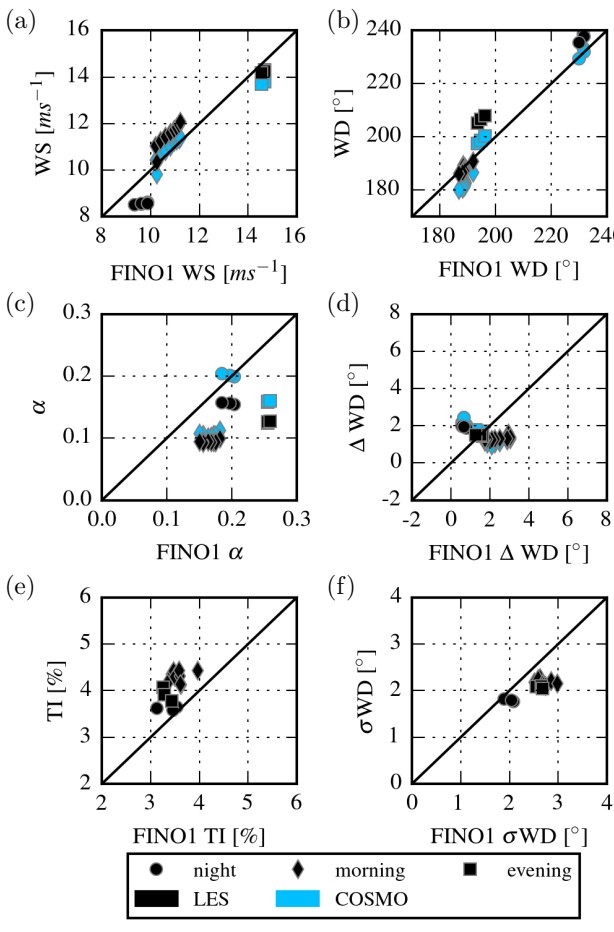

**Figure 8.** Comparison of the simulated state of the boundary layer with the measured state during the 15 ten-minute timeframes of the lidar measurements. Night period between 01:00 UTC and 01:40 UTC, morning period between 5:40 UTC and 7:30 UTC and evening period between 21:40 UTC and 22:10 UTC.





the simulations. Changes of atmospheric stability are small between the different times of measurements with the night and evening period in weekly stable conditions and the morning period in neutral conditions according to the classification in Peña et al. (2010), with the Monin-Obukhov length derived from the model fluxes.

**Figure 9.** Normalized wind fields from LES and lidar measurements. The third column shows horizontal cross sections along the lines at constant y, as depicted in the first two columns. The fourth column shows cross section at the same distances with the zero coordinate coinciding with the centerline of the wake. The full vertical distance between the horizontal lines in the cross section panels equals to a normalized velocity of one. The rows show ten-minute timeframes starting at (a) 1:30 UTC (b) 6:50 UTC (c) 21:40 UTC.




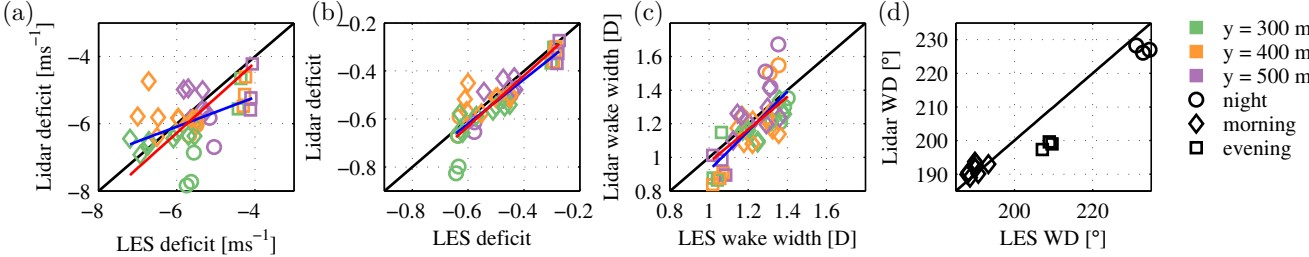

**Figure 10.** Scatter plots of the properties derived from the Gaussian-like fit to the wake profiles. The three different time intervals are marked with different markers as in Fig. 8. Colours represent the downstream distance from the turbine. Lines show simple linear regression (blue) and regression through the origin (red).

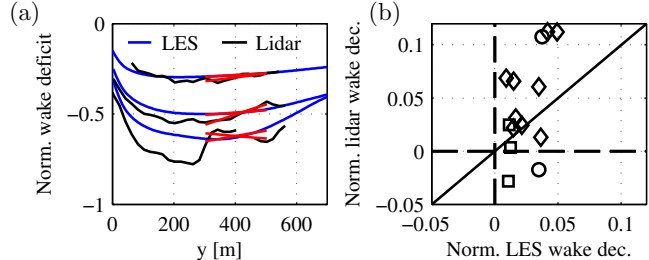

**Figure 11.** (a) Normalized wake deficit from the timeframes shown in Fig. 9 of LES and lidar measurements. Linear fit curve between y = 300 m and 500 m (red). (b) Comparison of the linear fit coefficients of the 15 timeframes to describe the decay of the wake deficit with distance.

For the wind turbine the three periods represent different operating conditions. With a rated wind speed of the turbine of 12.5 ms$^{-1}$, the wind speed range lies below rated wind speed during the night period. Below rated wind speed the turbine's power and thrust coefficient are nearly constant. During the evening period the wind speed is clearly above rated conditions, so the rotor speed is controlled by collective pitch movement of the blades, and the thrust coefficient decreases with increasing wind

5   speed. The morning period represents conditions that are around rated wind speed where the thrust coefficient should be lower and pitch control is occasionally applied.

Figure 9 shows ten-minute averages of the normalized hub-height wind speed during selected time intervals from simulation and measurements. For better comparison the LES results were averaged on cubes with a side length of 20 m centered at hub height, similar to the postprocessing of the lidar data as explained in Sec. 2.1. The slight disagreement of the inflow wind

10   speed was approached by normalizing the velocities of both flow fields. Lidar and LES wind speeds are normalized with the 90 m wind speed measured in the non-wake measurements or simulation data, respectively. To further remove the disagreement caused by the difference in wind direction, the flow fields were rotated in Fig. 9 (rightmost panels), so that the wake propagates along the y-axis.





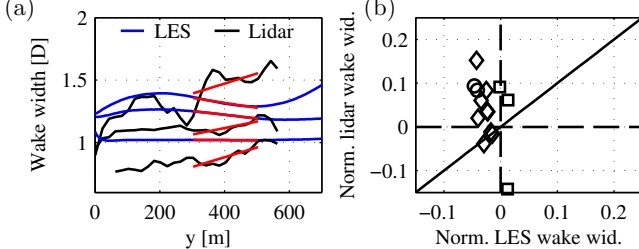

**Figure 12.** (a) 90th percentile width of the wake profiles as in Fig. 11(a). (b) Comparison of the linear fit coefficients of the 15 timeframes to describe the widening of the wake with distance.

The wake fields from the lidar measurements contain clearly still a lot of turbulent processes that are not removed by the ten-minute averaging. The reason is most probably the much lower sampling rate of the lidar measurements. Approximately 8000 single values contribute to the average on the 20 m grid in the LES, considering a timestep of about 2 Hz and the original grid resolution of 5 m. In contrast the sample velocities contributing to the lidar average vary from 100 to 350 individual line-

of-sight wind speed values and are not evenly distributed in space and time.

The results show that the unrotated wakes (Fig. 9 (third column)) match very well during the morning period where the wind direction appears to be nearly identical. In this period an asymmetry in the horizontal profile of the wake is also clearly visible, a phenomena related to the interaction of vertical wind shear with the rotation of the wake as shown in Vollmer et al. (2015). The amplitude of the wake deficit appears to be best simulated in the above-rated region in which the turbine operates during

the evening period. The low thrust leads to a wake that shows no signs of a double minima in the near wake as visible in the other measurements and simulations.

In the following an attempt is made to make a quantitative comparison between the measured and simulated wakes. Figure 10 shows wake statistics for all 15 time intervals derived from a Gaussian-like fit (Vollmer et al., 2016) to the wake profiles shown in Fig. 9(rightmost column). The Gaussian-like fit was found to be the most robust method to derive wake statistics, as

other tested methods struggle with the high degree of remaining turbulence in the lidar wind fields. Because the near wakes at y < 300 m deviate strongly from a Gaussian shape only statistics for larger distances are shown. The fit amplitude is defined as the wake deficit and the 90th percentile width of the fit as the wake width. Regression through the origin shows quite a good agreement between lidar and LES wake deficit, though the LES shows a tendency to simulate a higher wake deficit during the morning period and lower deficits otherwise. The spread decreases when normalizing the deficits. The slope of regression

between simulated and measured normalized deficits is 1.05. The simulated wakes are slightly wider in average, despite in below-rated conditions (night period), where the measured wakes are both wider and have a higher deficit.

For the analysis of the wake development with downstream advection, the extracted deficits and wake widths from the Gaussian-like function between y = 300 m and 500 m are fitted to linear functions. The resulting gradients are shown in Fig. 11 and 12. The spread of values from the lidar measurements is much larger, which we relate to the prevailing turbulent structures

in the lidar wind field. Interestingly, in the LES, the wake is not getting wider with distance in the first 500 m downstream





of the turbine as the widening takes place further downstream (Fig. 12). In the measurements, both cases, a widening and a narrowing of the wake are found.

## 5  Discussion

As shown in this paper the forcing of LES with mesoscale model input allows for time-dependent LES that change according to the synoptic meteorological conditions. We find however no proof that the LES can improve the quality of the comparison between the mesoscale model and measurements of the ambient wind profile. As the most important variables for wind farms are certainly a precise prediction of wind speed and direction, LES is not able to push these values closer to the measured values. Thus, for a good representation of the wind conditions the mesoscale model is the more crucial model.

The advantage of the presented methodology is that a transient state of the atmosphere can be used when comparing simulation and measurements. Thus, different situations during a continuous measurement campaign can be compared without deriving new boundary and initial conditions for each situation. In related publications e.g. Mirocha et al. (2015) a quasi static state was required to set up the simulations and the reproduction of the atmospheric background state was still found to be difficult.

A general problem of numerical simulations of high computational effort is the inclusion of horizontal inhomogeneities like orography or land-use. In the presented case we benefit from a well-developed boundary layer over a homogeneous sea surface. A simulation of a transient onshore boundary layer in complex terrain may require an approach with non-cyclic horizontal boundaries.

It becomes clear from our analysis that it is crucial for the ability to compare simulations with measurements that the measurement data is postprocessed for the specific purpose of validation. Here we were interested in the mean flow field at hub height and even during a relatively long time interval there were just a handful of measurements left for validation. Due to the lower sampling rate of the lidar measurements the available ten-minute averaged data still contains too many turbulent structures that make a comparison of derived wake statistics difficult. Thus, for a fair comparison of mean properties the sampling rate should be as high as possible or a smart long term averaging solution over similar conditions is necessary.

LES can deliver more information about the flow field than any combination of measurement devices. For the development and validation of lower-fidelity wind farm models this information can enhance the data pool of measurement campaigns and give insights into the physical reasons for the wake movement. With the ultimate goal to improve wind farm models used for wind farm layout design or wind farm control to reduce the cost of wind energy, the enhancement of measurement results with LES can be an important step.

## 6  Conclusions

In this paper we introduce and test a method to simulate a wind turbine wake in LES with realistic forcing derived from a mesoscale simulation. The comparison with met mast data shows that the model chain is able to reproduce mean and turbulent



quantities of the marine wind field during the analyzed two days with the main deviation being the vertical shear in the lower 100 m. A better mesoscale model is likely to improve the match between simulation and measurements. The wake simulations are compared to lidar measurements downstream of a turbine of *alpha ventus*. We get very good agreements during certain timeframes where wind direction and wake shape almost match perfectly. Measures to describe the downstream

wake development like the decay of the deficit match less well, primarily because the spread of the statistics derived from the measurements is much larger. We conclude that the limited data set of the lidar measurements and the still prevailing turbulent structures in the ten-minute averages of this data makes it difficult to validate the performance of the model. The presented method might not only be valuable for the comparison of simulations with measurement data but could be also applied to study wind turbine or wind farm control in changing wind conditions.

*Acknowledgements.*  The authors gratefully acknowledge the efforts of the Wind Energy Systems group of ForWind who carried out the lidar measurements at *alpha ventus*, including Jörge Schneemann, Davide Trabucchi, Juan-Jose Trujillo and Stephan Voß. The work presented in this study was funded by the national research projects ”GWWakes” and ”OWEA Loads” (FKZ 0325397A-B and 0325577B; Federal Ministry for Economic Affairs and Energy) and by the project "ventus efficiens" (ZN3024, ministry of science and culture of Lower Saxony). Computer resources have been partly provided by the North German Supercomputing Alliance (HLRN).

We thank Deutscher Wetterdienst (DWD) for providing analysis data.



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
