# Peer review of "Transient LES of an offshore wind turbine"

_Wind Energy Science, 2017_

## Referee Comment (RC1) · Anonymous Referee #1 · 21 Jun 2017

General Comments:

This paper addresses the incorporation of mesoscale model output into a large-eddy simulation (LES) using cyclic (periodic) lateral boundary conditions. The LES wind speed, wind direction, and potential temperature fields are guided toward corresponding mesoscale simulated values, via a combination of advections, geostrophic wind forcing, and relaxation, allowing the LES to track some of the low-frequency variability captured within the mesoscale solution. Comparison of simulated and observed meteorological parameters indicates generally good agreement between the mesoscale and LES simulated fields, as well as between the simulations and observations, on timescales of a few hours or greater. The emphasis then switches to comparing simulated turbine wakes, modeled using an actuator disk with rotation, within an LES forced with mesoscale input, to lidar-observed wakes from an offshore wind farm. The simulations are shown to capture observed bulk wake properties reasonably well in the

aggregate, with success quantified mostly using best fit parameters to a Gaussian wake model.

The examination demonstrates both the successes and the limitations of the proposed simulation framework; the LES closely follows the mesoscale wind speed and direction profiles, as well as changes of temperature, however is not capable of improving bulk vertical wind shear beyond the mesoscale simulations, and fails to reproduce some observed wake properties.

While starting with a promising and interesting premise, the value of the study is compromised by both some methodological shortcomings, and, as importantly, by Discussion and Conclusions sections that fail to engage interesting components of the study, instead proceeding with an unsubstantiated overstatement of the utility of LES, before transitioning into a desultory presentation of the general difficulties of high-fidelity simulation, and comparison of simulations and observations. These could be the most informative and illuminating sections of the paper, especially when a new and well motivated technique is examined, such as is the case here. Instead, the paper ends up feeling unfinished, with much potentially useful discussion omitted.

A key objection is the assertion that LES does not improve representation of the ambient wind field beyond a mesoscale model, based upon the observation that the LES is not able to "push" simulated parameter values from the mesoscale simulated values closer to the observations. First, there is an insufficient basis from this small study to generalize about the ability of LES to improve upon a mesoscale prediction. Further, any such "improvement" depends upon the desired quantity. While wind speed and direction are certainly crucial, turbulence quantities are also important, influencing power production, stress loading and wake evolution, for example. If done correctly, LES can provide good representations of these characteristics. However, the expectation of the LES to "push" certain variables closer to observed values than as represented within the mesoscale simulation is a bit misplaced, especially given that i) the LES herein was forced toward those mesoscale values and ii) the nearly steady and homogeneous conditions simulated herein are precisely the conditions for which mesoscale turbulence parameterizations would be expected to function quite well.

A more reasonable expectation, in my view, would be that the LES could resolve the classical turbulence spectrum consistent with the slowly varying flow component as simulated by the mesoscale model. I would be interested to know how well the LES met this more reasonable expectation.

So, how did the LES conducted herein perform in that respect? Such was not a central inquiry of the present study, however some hints were provided. These limited results lead me to question if the LES conducted herein were somehow deleteriously impacted by the incorporation of the mesoscale forcing. Evidence for this hypothesis includes i) the much lower magnitude of the ten-minute simulated variability, relative to that which was observed, shown in Fig. 7, ii) the smaller variances shown in Table 1, especially under the influence of the mesoscale forcing—note how much larger the variances are when u-advection is ignored, and iii) the absence of variability in the background simulated flow, as well as symmetry of the wake structure, relative to the observations, shown in Fig. 9.

These questions can be answered via more substantial assessment of the LES flow field, which is my key recommendation. At a minimum, some comparison of simulated and observed spectra and stresses should be carried out if possible, and if not, at least spectra and/or stress profiles from the simulations should be presented and compared with the results of other studies. Only after establishing that the LES is capturing the classical energy spectrum well can assessment of its applicability be undertaken.

Following that, I think a more comprehensive examination of the wakes would also strengthen the paper. While the formal quantitative comparison is restricted to the portions of the wake for which the Gaussian model can function, other aspects of the wakes (far wake, meander, etc.) could be discussed at least qualitatively.

These examinations could lead to a much more illuminating discussion of both the

promise and the difficulties regarding the application of mesoscale information into quasi-idealized LES with cyclic boundary conditions, an interesting and timely topic that deserves careful examination. This paper represents a good first step in that direction that, with some polishing, could be a very useful contribution to the literature.

Below are a handful of minor corrections and additional suggestions:

P1, L14: replace first occurrence of "of" with "to". P1, L15: which has "been" established... P1, L17: Sentence beginning on this line. Please describe briefly some of the errors and why those are so large. P1, L20: Due to the "generally" lower... "frequently" more persistent... P1, L22: Stable conditions are not unique to offshore environments; onshore sites typically feature stronger static stability due to more rapid nighttime cooling over land than water. P1, L25: Please add "simplified" or "Fast running" to the sentence beginning on this line, as there is a wide range of "engineering" models, some of which are very high fidelity and therefore too slow to be used in the described capacity. P2, L5: please remove "exemplary". P2, L8: please remove "permanent". P2, L13: replace "fair" with "meaningful". P2, L19: replace "a lot of" with "many". P2, L25: replace "us to include" with "for inclusion of". P2, L34: replace "wind turbine" with "actuator". P3, L20: Either include enough detail about precisely what is meant by "enough" and "a lot of" so that another researcher may duplicate your data processing methodology. P3, Eq. 1. Since turbulence closure is an important aspect of LES, please describe the approach utilized herein. P6, L11: data is "are" averaged. P6, Eq. 3: Please define f3. P7, L1: density and pressure pressure and density, respectively. P7, L10: are close to agree well with. P9, Table 1: I am not able to understand this table. First, why would statistics of the measurements (F1) be different for different model configurations (rows)? More explanation would help clarify. Second, why were sigma wd and sigma ws so much larger when momentum advection was turned off? This is potentially important. It seems this might be doing something significant within the LES. I think looking at spectra, for example, could provide some insight. P10, Fig. 7 caption: Is the black line the hourly average, and the gray line the ten-minute

average? Also, the caption claims that the power law coefficient is defined in the text but I could not find that. P11, Sentence beginning on Line 2: I do not agree that the ten-minute variability is well reproduced by the models. The model parameter values appear to exhibit significantly less variability than the data. P11, L23: What are the constant values of the drag coefficients used for the nacelle and tower? P13, Fig. 9: Seems to be much more variability in observed than LES background. Perhaps this is important in wake spreading? Also, how about showing more of the far wake regions? Even if analysis is restricted to 3-5 D due to the wake recognition algorithm, it would be nice to see how the far wake widens and dissipates in the simulations relative to the observations. P15, L1: Please replace "a lot" with something more specific. P15, L10. Any speculation on why the thrust coefficient so much lower in the operating lidar than in the simulation, if I am understanding correctly? P15, L14: Space between 9 and (. P15, L18: Please explain why you think the LES has these biases? Higher deficit in the morning, morning, lower other times. P15, L20: despite "being operated" in ... End of page 15: I think more discussion/analysis of the wake widening would be helpful. Is the simulation perhaps not capturing some interesting physical process, such as maybe hub vortex shedding, which leads to widening/meandering of the near wake?

---

## Referee Comment (RC2) · Anonymous Referee #2 · 3 Jul 2017

**Review of the manuscript, titled "*Transient LES of an offshore wind turbine*"**

In this manuscript, a methodology is presented, in which a large-eddy simulation (LES) model is combined with a numerical weather prediction (NWP) model to simulate the flow through an offshore wind farm. The results of this combination of models are compared with field measurements (both lidar and met mast) in an offshore wind farm in the North Sea. The topic is interesting and it is a direction, in which wind energy community should/will eventually move. Nevertheless, there are some issues that the authors should address, before this manuscript can officially get published. These issues are listed below:

- The statement that the authors have made about LES in their discussion section (Page 16, L 5-8) is very premature and too generalizing. The only thing one can conclude about the comparison of the LES results and the measurements reported in this paper (in its current form) is: the type of LES used in this paper and the methodology used in this paper to combine LES and NWP and the way LES is set up in this paper and the way LES was run in this paper and the way the results (both LES and measurements) were post-processed in this paper, resulted in the comparison reported in this paper. In fact, the accuracy of LES in prediction of wind-turbine wakes has been well tested and proved to be satisfactory in the literature. I think, with respect to these results, the authors should explain and explore what has gone wrong or what the inherent limitations are, rather than simply stating "*We find however no proof that the LES can improve the quality of the comparison…*" or "*…LES is not able to push these values closer to the measured values.*"

  Here, another question comes to one's mind: have the authors tested and validated their LES for flow through wind turbines against more controlled and classical wind-tunnel experiments (which exists in the literature)? (if yes, please mention the reference in the text) It is very important that authors first make sure their LES model combined with their actuator disk model works well, and only then they can aim to test the accuracy of LES in such a complicated case that is described in this paper. In other words, the steps should be taken one by one.

- Regarding the inflow boundary condition the authors have written:
  " *… we made simulations without the turbine in parallel and simulated only intervals of 30 min maximum with wind turbine with a 3 min precursor phase for the development of the wake.*"
  It is clear that the authors have used a precursor simulation to generate the inflow; however, it is very vague how they have done it. Please state very clearly how you have performed your precursory simulation and how you have linked your precursory simulation to your main simulation. Have you used a buffer/fringe zone to overcome the periodic boundary condition? How have implemented it? In validation of LES results against measurements, it is well known that, an appropriate inflow is crucial. For example, one can consider this quote from Andreas Kempf:
  "*In general, LES inflow-conditions involve far more detail than those for a RANS, and if the LES is not provided with this data, the results cannot be expected to be superior to RANS results.*"[1]
* * *
[1] Kempf, Andreas M. "LES validation from experiments." *Flow, Turbulence and Combustion* 80.3 (2008): 351-373.

- In your LES formulation (Eq. 1), please indicate the term responsible for the SGS stresses. Is "kinematic viscosity of momentum" the turbulent viscosity?

- In Page 10, Lines 5-10, you have discussed about the choice of the relaxation time. You have finally chosen tau=4 h. You have first mentioned that the errors with respect to FINO1 measurements are not a good criterion to assess the value of tau, and then you have said that the choice of tau was a qualitative decision. Please substantiate the choice of tau. Here it seems tau=1.5 h is a better choice based on the errors, and with tau=48 h "LES boundary layer develops more independently". So what was your criterion/criteria for choosing tau=4 h?

- Define exactly and mathematically the "wake width" and "wake deficit". Explain how you have calculated these quantities both for lidar data and for LES data.

- The correspondence of Fig. 12 and Fig. 9 seem questionable. For example, based on Fig. 12 the width of the wake measured by lidar in the evening period has doubled from y=200 m to y=500 m; however, one cannot see such an increase in Fig. 9. Moreover, in the night period, based on Fig. 12 we have a quick 0.5D jump in the wake width between y=300 and 400 (for the lidar data); this jump is again not clear in Fig. 9. It seems to me a bit of inconsistency. Furthermore, the differences in the wake width in Fig. 9 (between LES and lidar) does not seem as large as in Fig. 12.

- In Fig. 10, the difference between panel (a) and (b) is not clear. It is not mentioned in the caption, and it is not even discussed in the text.

- Please indicate in Fig. 1 the position of the met mast "FINO1". In the same figure, please indicate the North direction.

- Some editorial comments:
    - Make a distinction between RMSE symbol used in Table 1 and the RMS of fluctuations used in the rest of the paper (e.g. Fig. 7, 8, etc.).
    - Define all the abbreviations the first time you use them in the text. For example: DEWI, PPI, PALM.
    - "u_g" in Eq. 1 has the "LS" subscript, but in Eq. 3 does not. Are they the same variables? If yes, use a consistent notation for both.
    - Page 5, Line 10: I think you need a comma after "Compared to onshore"
    - Page 2, Line 3: "are" should be changed to "is".

---

## Author Comment (AC2) · 31 Jul 2017

**1 Response to WES2017-16 RC2**

The authors thank the referee for the helpful review. The list of issues that were brought up by the referee is addressed in the following.

**General comments**

In this manuscript, a methodology is presented, in which a large-eddy simulation (LES) model is combined with a numerical weather prediction (NWP) model to simulate the flow through an offshore wind farm. The results of this combination of models are compared with field measurements (both lidar and met mast) in an offshore wind farm in the North Sea. The topic is interesting and it is a direction, in which wind energy community should/will eventually move. Nevertheless, there are some issues that the authors should address, before this manuscript can officially get published. These issues are listed below:

**Comment 1**

The statement that the authors have made about LES in their discussion section (Page 16, L 5-8) is very premature and too generalizing. The only thing one can conclude about the comparison of the LES results and the measurements reported in this paper (in its current form) is: the type of LES used in this paper and the methodology used in this paper to combine LES and NWP and the way LES is set up in this paper and the way LES was run in this paper and the way the results (both LES and measurements) were post-processed in this paper, resulted in the comparison reported in this paper. In fact, the accuracy of LES in prediction of wind-turbine wakes has been well tested and proved to be satisfactory in the literature. I think, with respect to these results, the authors should explain and explore what has gone wrong or what the inherent limitations are, rather than simply stating We find however no proof that the LES can improve the quality of the comparison... or ...LES is not able to push these values closer to the measured values. Here, another question comes to ones mind: have the authors tested and validated their LES for flow through wind turbines against more controlled and classical wind-tunnel experiments (which exists in the literature)? (if yes, please mention the reference in the text) It is very important that authors first make sure their LES model combined with their actuator disk model works well, and only then they can aim to test the accuracy of LES in such a complicated case that is described in this paper. In other words, the steps should be taken one by one.

**Authors' comment:** *Thanks for this helpful comment. Before answering in more detail, we would like to clarify that the objective of the paper is to introduce and test the methodology of LES driven by mesoscale model input for wind turbine wake modelling. In the scope of this manuscript we further compare full scale measurements of wind turbine wakes to simulations conducted with the model chain. As mentioned by the referee later in the review, the challenge with LES is to use the right inflow and boundary conditions. The concept of the manuscript is to show that the presented way of using mesoscale model output can serve as a good approximation of these conditions.*

*We agree with the referee that, in general, it has been shown that LES of wind turbine models are capable to reproduce wake characteristics. Full-sized wind turbines in the atmospheric*

*boundary layer (ABL), however, face a wide range of wind conditions that strongly influence the wake characteristics.Replicating the wind conditions for non-neutral conditions is challenging with idealized quasi-static LES, as the atmospheric boundary layer is never in a steady state. In our view, there is no consistent work yet that proposes how to derive the boundary conditions for LES of this wide range of measured wind conditions.*

*Regarding the accuracy of the LES in more controlled conditions, we compared the thrust and power curves of the simulated wind turbine with the confidential curves given by the constructor in idealized LES runs. For the modelled wind speeds both curves are almost equal.*

*Considering the phrase cited by the referee, that "We find however no proof that the LES can improve the quality of the comparison...": This phrase is only related to the mean ambient wind field created by the LES in comparison to the numerical weather prediction model. Here we find no proof, that the mean wind field and measures of the mean wind profile from the LES are closer to the measured wind field. We will add that the study only looks at a very short time horizon and only near-neutral stability conditions and is thus not very representative. For more discussion of the topic we delegate the reader to the authors' response on comment 1 in the answer to referee RC1.*

**Comment 2**

Regarding the inflow boundary condition the authors have written: ... we made simulations without the turbine in parallel and simulated only intervals of 30 min maximum with wind turbine with a 3 min precursor phase for the development of the wake. It is clear that the authors have used a precursor simulation to generate the inflow; however, it is very vague how they have done it. Please state very clearly how you have performed your precursory simulation and how you have linked your precursory simulation to your main simulation. Have you used a buffer/fringe zone to overcome the periodic boundary condition? How have implemented it? In validation of LES results against measurements, it is well known that, an appropriate inflow is crucial. For example, one can consider this quote from Andreas Kempf: In general, LES inflow-conditions involve far more detail than those for a RANS, and if the LES is not provided with this data, the results cannot be expected to be superior to RANS results.

**Authors' comment:** *The description of the procedure we use might be a bit too complicated in the manuscript. We will add a more detailed step by step explanation of the procedure. What we do is to write out binary data of the wind fields from the simulations without turbines at the time steps at which we want to start the turbine simulations. These data serve as our base of precursor runs. The turbine simulations are then run with the same boundary conditions as the simulations without turbine, just with the added body forces of the wind turbine. Due to the cyclic horizontal boundary conditions, the changes to the wind field from the turbine wake continue to propagate inside the domain. We benefit from the wind direction that is never directly along one of the main axis of the model during the turbine simulations. The wake thus never interacts with the turbine already during the first flow through the model. As there is still a change to the boundary layer turbulence induced by the wake, the turbine simulations are restricted to 30 min. A buffer/fringe zone might overcome this problem, however the continuously changing boundary conditions make the implementation of this zone a complicated task.*

*Regarding the quote by Prof Kempf we refer to the answer on the previous comment.*

**Comment 3**

In your LES formulation (Eq. 1), please indicate the term responsible for the SGS stresses. Is kinematic viscosity of momentum the turbulent viscosity?

**Authors' comment:** *For simplicity this equation represent the Navier-Stokes equation of momentum before any approximation. For the final set of equations the Boussinesq-approximation and Reynolds-averaging are applied. The terms that represents the sub-gridscale fluxes has its origin in term 1 of the shown equation. In the final equations the molecular diffusivity term with the kinematic viscosity is actually neglected, because it is several orders of magnitude smaller compared to the turbulent viscosity. For the SGS terms a 1.5 order closure after [1] is used. For more information about the model we like to refer to [2].*

**Comment 4**

In Page 10, Lines 5-10, you have discussed about the choice of the relaxation time. You have finally chosen tau=4 h. You have first mentioned that the errors with respect to FINO1 measurements are not a good criterion to assess the value of tau, and then you have said that the choice of tau was a qualitative decision. Please substantiate the choice of tau. Here it seems tau=1.5 h is a better choice based on the errors, and with tau=48 h LES boundary layer develops more independently. So what was your criterion/criteria for choosing tau=4 h?

**Authors' comment:** *Here we also follow the recommendation of Schalkwijk et al.[3] and Heinze et al.[4] as it is not straightforward to conclude from the comparison between the mean wind field in the simulation and the measurements, which setup is the best. In the manuscript we chose the differences in wind speed and wind direction near hub height as measures for the error. The sources of errors can come from multiple sources, e.g. from a difference in magnitude and phase of the mesoscale model, that is not able to model the scenario, from the higher fluctuations of the mean values in the measurements or from the inertia of the flow in the LES domain that prevents the model to follow the changing background conditions. A lower relaxation time only improves the last aspect. On the other hand it forces the LES profile towards the mesoscale profile, which is not necessarily correct. If the LES has more freedom to develop it might improve the representation of the wind profile. While this might not be the case in the present study, we showed this recently in a proceedings paper [5].*

**Comment 5 & Comment 6**

Define exactly and mathematically the wake width and wake deficit. Explain how you have calculated these quantities both for lidar data and for LES data.

The correspondence of Fig. 12 and Fig. 9 seem questionable. For example, based on Fig. 12 the width of the wake measured by lidar in the evening period has doubled from y=200 m to y=500 m; however, one cannot see such an increase in Fig. 9. Moreover, in the night period, based on Fig. 12 we have a quick 0.5D jump in the wake width between y=300 and 400 (for the lidar data); this jump is again not clear in Fig. 9. It seems to me a bit of inconsistency.

[Figure]

Figure 1: Wind field at 1:30 UTC. (a) LES wake profiles (dashed) and fits to the profiles. (b) Lidar wake profiles (dashed) and fits. (c) Fitted profiles from LES (blue) and Lidar (black). (d) Normalized deficit and wake width from LES (blue) and Lidar (black).

Furthermore, the differences in the wake width in Fig. 9 (between LES and lidar) does not seem as large as in Fig. 12.

**Authors' comment:** *We use a Gaussian fit like in [6]. We describe the parameters in p.15 L.17: "The fit amplitude is defined as the wake deficit and the 90th percentile width of the fit as the wake width." In reaction to the review we modified the equation to include the near wake. The fitting function now consists of two overlayed Gaussian-like functions and is given by Eq. 1.*

$$u_{def}(x) = b \, \exp\left(-\left(\frac{x-c}{d}\right)^2\right) - e \, b \, \exp\left(-\left(\frac{x-c}{f \, d}\right)^2\right) \tag{1}$$

*With this function, the area of lower deficit in the near wake is represented by the second exponential function, with $e,f \in [0,1]$. The wake width (90th percentile) we get from this fit is $L_{def} = \sqrt{2} \cdot 1.64 \cdot d$, as deficit we chose the minimum value of each fitted $u_{def}$. In the revised manuscript we will modify the figures about wake decay and width and add figures like Fig.1 to show how the parameters were derived. The fit to the measured data still delivers quite noisy trajectories, which we discussed in the Discussion section of the manuscript.*

**Comment 7**

In Fig. 10, the difference between panel (a) and (b) is not clear. It is not mentioned in the caption, and it is not even discussed in the text.

**Authors' comment:** *The difference is simply that panel (b) shows the deficit normalized by the ambient flow field as in Fig. 9. We will add a description.*

**Comment 8**

Please indicate in Fig. 1 the position of the met mast FINO1. In the same figure, please indicate the North direction.

**Authors' comment:** *Will be added.*

**Comment 9: Editorial comments**

Some editorial comments:

- Make a distinction between RMSE symbol used in Table 1 and the RMS of fluctuations used in the rest of the paper (e.g. Fig. 7, 8, etc.).

- Define all the abbreviations the first time you use them in the text. For example: DEWI, PPI, PALM.

- $u_g$ in Eq. 1 has the LS subscript, but in Eq. 3 does not. Are they the same variables? If yes, use a consistent notation for both.

- Page 5, Line 10: I think you need a comma after Compared to onshore

- Page 2, Line 3: are should be changed to is.

**Authors' comment:** *We will correct or edit the manuscript based on the comments*

**References**

[1] Deardorff J 1980 *Boundary-Lay. Meteorol.* **18** 495–527

[2] Maronga B, Gryschka M, Heinze R, Hoffmann F, Kanani-Sühring F, Keck M, Ketelsen K, Letzel M O, Sühring M and Raasch S 2015 *Geosci. Model Dev.* **8** 2515–2551

[3] Schalkwijk J, Jonker H J J, Siebesma A P and Bosveld F C 2015 *Monthly Weather Review* **143** 828–844 (*Preprint* http://dx.doi.org/10.1175/MWR-D-14-00293.1)

[4] Heinze R, Moseley C, Böske L N, Muppa S, Maurer V, Raasch S and Stevens B 2016 *Atmospheric Chemistry and Physics Discussions* **2016** 1–37

[5] Vollmer L, Lee J C Y, Steinfeld G and Lundquist J K 2017 *Journal of Physics: Conference Series* **854** 012050

[6] Vollmer L, Steinfeld G, Heinemann D and Kühn M 2016 *Wind Energy Science* **1** 129–141

---

## Author Comment (AC1)

**1 Response to WES2017-16 RC1**

The authors thank the referee for the helpful review. We tried to identify the main requests to the manuscript from the text and address them in the following.

**General comments**

This paper addresses the incorporation of mesoscale model output into a large-eddy simulation (LES) using cyclic (periodic) lateral boundary conditions. The LES wind speed, wind direction, and potential temperature fields are guided toward corresponding mesoscale simulated values, via a combination of advections, geostrophic wind forcing, and relaxation, allowing the LES to track some of the low-frequency variability captured within the mesoscale solution. Comparison of simulated and observed meteorological parameters indicates generally good agreement between the mesoscale and LES simulated fields, as well as between the simulations and observations, on timescales of a few hours or greater. The emphasis then switches to comparing simulated turbine wakes, modeled using an actuator disk with rotation, within an LES forced with mesoscale input, to lidar-observed wakes from an offshore wind farm. The simulations are shown to capture observed bulk wake properties reasonably well in the aggregate, with success quantified mostly using best fit parameters to a Gaussian wake model. The examination demonstrates both the successes and the limitations of the proposed simulation framework; the LES closely follows the mesoscale wind speed and direction profiles, as well as changes of temperature, however is not capable of improving bulk vertical wind shear beyond the mesoscale simulations, and fails to reproduce some observed wake properties. While starting with a promising and interesting premise, the value of the study is compromised by both some methodological shortcomings, and, as importantly, by Discussion and Conclusions sections that fail to engage interesting components of the study, instead proceeding with an unsubstantiated overstatement of the utility of LES, before transitioning into a desultory presentation of the general difficulties of high-fidelity simulation, and comparison of simulations and observations. These could be the most informative and illuminating sections of the paper, especially when a new and well motivated technique is examined, such as is the case here. Instead, the paper ends up feeling unfinished, with much potentially useful discussion omitted.

**Authors' comment:** *The authors will revise the Discussion and Conclusion sections to focus on the aspects mentioned in the review. Regarding the methodological shortcomings mentioned by the referee, we could not identify which shortcomings were ment by reading the review. We however assume that this comment is related to the requested additional analysis in the following comments and a more profound discussion of the model chain ability to reproduce the ambient wind and the wake.*

**Comment 1: Evaluation of the representation of the ambient wind field**

A key objection is the assertion that LES does not improve representation of the ambient wind field beyond a mesoscale model, based upon the observation that the LES is not able to push simulated parameter values from the mesoscale simulated values closer to the observations. First, there is an insufficient basis from this small study to generalize about the ability of LES to improve upon a mesoscale prediction. Further, any such improvement depends upon the desired quantity. While wind speed and direction are certainly crucial, turbulence quantities are also important, influencing power production, stress loading and wake evolution, for example. If done correctly, LES can provide good representations of these characteristics. However, the expectation of the LES to push certain variables closer to observed values than as represented within the mesoscale simulation is a bit misplaced, especially given that i) the LES herein was forced toward those mesoscale values and ii) the nearly steady and homogeneous conditions simulated herein are precisely the conditions for which mesoscale turbulence parameterizations would be expected to function quite well. A more reasonable expectation, in my view, would be that the LES could resolve the classical turbulence spectrum consistent with the slowly varying flow component as simulated by the mesoscale model. I would be interested to know how well the LES met this more reasonable expectation. So, how did the LES conducted herein perform in that respect? Such was not a central inquiry of the present study, however some hints were provided. These limited results lead me to question if the LES conducted herein were somehow deleteriously impacted by the incorporation of the mesoscale forcing. Evidence for this hypothesis includes i) the much lower magnitude of the ten-minute simulated variability, relative to that which was observed, shown in Fig. 7, ii) the smaller variances shown in Table 1, especially under the influence of the mesoscale forcingA Tnote how much larger the variances are when u-advection is ignored, and iii) the absence of variability in the background simulated flow, as well as symmetry of the wake structure, relative to the observations, shown in Fig. 9. These questions can be answered via more substantial assessment of the LES flow field, which is my key recommendation. At a minimum, some comparison of simulated and observed spectra and stresses should be carried out if possible, and if not, at least spectra and/or stress profiles from the simulations should be presented and compared with the results of other studies. Only after establishing that the LES is capturing the classical energy spectrum well can assessment of its applicability be undertaken.

P11, Sentence beginning on Line 2: I do not agree that the ten-minute variability is well reproduced by the models. The model parameter values appear to exhibit significantly less variability than the data.

**Authors' comment:**  *The referee rightfully states that the present study does not present a sufficient basis to generalize about the ability of LES to improve upon a mesoscale prediction. The key recommendation is to perform a more substantial assessment of the ambient flow conditions with a focus on the resolved turbulence frequencies.*

*To address this recommendation we propose to include the following aspects in the revised manuscript. Firstly, a comparison of the observed and simulated spectra for the period of the two simulated days (Fig. 1). As shown in the figure the LES model shows the typical stronger falloff in the high frequency domain, related to the cut-off of the highest frequencies by the implicit filtering of sub-grid frequencies. As stated in the manuscript, the variance on the 10 min scale is nearly preserved. At lower frequencies the LES spectra first falls off, which is not the case for the measurement. This gap in the time period ranges from about 0.5 to 12 h. To be able to reproduce the frequencies in this range, the horizontal extension of the LES model has to be much larger at least [1]. The spectra from the mesoscale model is not shown in Fig. 1, because we use only hourly resolved data. However, Vincent et al.[2] i.a. have shown that mesoscale models are not able to resolve most of the measured fluctuations in the multiple hour range. For the discussion of the general ability of the model chain to model the*

[Figure]

Figure 1: Power Spectral density from the LES (blue) and from the 1 Hz cup measurements at 90 m at FINO1 with different window sizes over the two day period. $2^{15}$ $s$ (grey). $2^{10}$ $s$ (black). Black isolated line depicts the slope from Kolmogorov cascade.

*turbulence spectra we will include the findings of [1, 3] in the discussion section and compare the approach with the nesting approach discussed in Munoz-Esparza et al.[4].*

*Considering the above-mentioned, we still find no proof in this study that the modelling of the mean wind profile is improved by the downscaling with LES. Of course, LES introduces turbulence quantities that are crucial for the wake development, which is the main motivation of the research presented in this paper. The improvement of mesoscale models by using LES on the other hand is not a motivation for this specific paper and we will at least weaken the statement in the revised manuscript.*

**Comment 2: A more comprehensive examination of the wakes**

Following that, I think a more comprehensive examination of the wakes would also strengthen the paper. While the formal quantitative comparison is restricted to the portions of the wake for which the Gaussian model can function, other aspects of the wakes (far wake, meander, etc.) could be discussed at least qualitatively. These examinations could lead to a much more illuminating discussion of both the promise and the difficulties regarding the application of mesoscale information into quasi-idealized LES with cyclic boundary conditions, an interesting and timely topic that deserves careful examination. This paper represents a good first step in that direction that, with some polishing, could be a very useful contribution to the literature.

I think more discussion/analysis of the wake widening would be helpful. Is the simulation perhaps not capturing some interesting physical process, such as maybe hub vortex shedding, which leads to widening/meandering of the near wake?

P13, Fig. 9: Seems to be much more variability in observed than LES background. Perhaps this is important in wake spreading? Also, how about showing more of the far wake

regions? Even if analysis is restricted to 3-5 D due to the wake recognition algorithm, it would be nice to see how the far wake widens and dissipates in the simulations relative to the observations.

**Authors' comment:** *We agree with the referee that a examination of other aspects of the wakes, e.g. far wake, meandering would be a very interesting topic. In this manuscript we are, however, restricted to the available measurements, which were performed with the focus on measuring the mean wake profile at hub height. Range and sampling rate of the measurements are too low to study far wake or wake meandering, respectively. We expect that the presented methodology allows to do such a comparison with measurements in changing atmospheric stability conditions, that are believed to be the main reason for different observed meandering behaviour. We suggest to add a paragraph on this topic in the discussion.*

**Comment 3: Table 1**

P9, Table 1: I am not able to understand this table. First, why would statistics of the measurements (F1) be different for different model configurations (rows)? More explanation would help clarify.

**Authors' comment:** *We will focus a bit more on describing the table. What is shown in the columns and termed $\sigma X$ is not the changing statistics of the measurements but the statistics of the difference between model result and measurement that change for every model setup. As the notation might be a bit missleading, we will exchange it.*

**Comment 4: Momentum advection**

Second, why were sigma wd and sigma ws so much larger when momentum advection was turned off? This is potentially important. It seems this might be doing something significant within the LES. I think looking at spectra, for example, could provide some insight.

**Authors' comment:** *Without momentum advection, the inertia of the flow is too high to follow the trend given by the mesoscale model. The nudging term is already added to overcome this inertia, but momentum advection appears to further decrease the difference to the input trend and the measurements. Figure 2 shows that the mean wind speed without momentum advection is much higher on the second day between 12 and 18 UTC. Here the flow is still reacting on the sudden increase of the geostrophic wind speed (large scale pressure gradient) around 6 UTC (see Fig.5d), naturally resulting in an oscillation of the flow in the domain. The sink of momentum by the momentum advection terms as shown in Fig. 5h dampens this oscillation. We will add this figure for clarification.*

**Minor comments**

P1, L14: replace first occurrence of of with to. P1, L15: which has been established. . . P1, L17: Sentence beginning on this line. Please describe briefly some of the errors and why those are so large. P1, L20: Due to the generally lower. . . frequently more persistent. . . P1, L22: Stable conditions are not unique to offshore environments; onshore sites typically feature stronger static stability due to more rapid nighttime cooling over land than water. P1, L25: Please add simplified or Fast running to the sentence beginning on this line, as there is a wide

[Figure]

Figure 2: Comparison of 70 m wind speed from different model setups of PALM, the mesoscale model and from the measurements at FINO1.

range of engineering models, some of which are very high fidelity and therefore too slow to be used in the described capacity. P2, L5: please remove exemplary. P2, L8: please remove permanent. P2, L13: replace fair with meaningful. P2, L19: replace a lot of with many. P2, L25: replace us to include with for inclusion of. P2, L34: replace wind turbine with actuator. P3, L20: Either include enough detail about precisely what is meant by enough and a lot of so that another researcher may duplicate your data processing methodology. P3, Eq. 1. Since turbulence closure is an important aspect of LES, please describe the approach utilized herein. P6, L11: data is are averaged. P6, Eq. 3: Please define f3. P7, L1: density and pressure pressure and density, respectively. P7, L10: are close to agree well with. P10, Fig. 7 caption: Is the black line the hourly average, and the gray line the ten-minute average? Also, the caption claims that the power law coefficient is defined in the text but I could not find that. P11, L23: What are the constant values of the drag coefficients used for the nacelle and tower? P15, L1: Please replace a lot with something more specific. P15, L10. Any speculation on why the thrust coefficient so much lower in the operating lidar than in the simulation, if I am understanding correctly? P15, L14: Space between 9 and (. P15, L18: Please explain why you think the LES has these biases? Higher deficit in the morning, morning, lower other times. P15, L20: despite being operated in . . .

**Authors' comment:** *We will address the remarks in the revised manuscript.*

**References**

[1] Schalkwijk J, Jonker H J J, Siebesma A P and Bosveld F C 2015 *Monthly Weather Review* **143** 828–844 (*Preprint* http://dx.doi.org/10.1175/MWR-D-14-00293.1)

[2] Vincent C, Larsén X, Larsen S and Sørensen P 2013 *Boundary-Layer Meteorology* **2** 297–318

[3] Heinze R, Moseley C, Böske L N, Muppa S, Maurer V, Raasch S and Stevens B 2016 *Atmospheric Chemistry and Physics Discussions* **2016** 1–37

[4] Muñoz-Esparza D, Kosovic B, Mirocha J and van Beeck J 2014 *Boundary-Layer Meteorol.* **153** 409–440

---

## Author Response (AR1)

**Response to Editor**

Dear Luciano Castillo,

the attached manuscript was revised considering the author's answers to the two referees as well as your request for further minor corrections. The mayor changes to the previous document are marked in italics.

Kind Regards,

[revised manuscript text omitted]